# Chronic Exposure to TDI Induces Cell Migration and Invasion via TGF-β1 Signal Transduction

**DOI:** 10.3390/ijms24076157

**Published:** 2023-03-24

**Authors:** Dong-Hee Han, Min Kyoung Shin, Jin Wook Oh, Junha Lee, Jung-Suk Sung, Min Kim

**Affiliations:** Department of Life Science, Dongguk University-Seoul, Biomedi Campus, 32 Dongguk-ro, Ilsandong-gu, Goyang 10326, Gyeonggi-do, Republic of Korea

**Keywords:** toluene diisocyanate, transforming growth factor-beta1, chronic exposure, epithelial-mesenchymal transition, migration, invasion

## Abstract

Toluene diisocyanate (TDI) is commonly used in manufacturing, and it is highly reactive and causes respiratory damage. This study aims to identify the mechanism of tumorigenesis in bronchial epithelial cells induced by chronic TDI exposure. In addition, transcriptome analysis results confirmed that TDI increases transforming growth factor-beta 1 (TGF-β1) expression and regulates genes associated with cancerous characteristics in bronchial cells. Our chronically TDI-exposed model exhibited elongated spindle-like morphology, a mesenchymal characteristic. Epithelial-mesenchymal transition (EMT) was evaluated following chronic TDI exposure, and EMT biomarkers increased concentration-dependently. Furthermore, our results indicated diminished cell adhesion molecules and intensified cell migration and invasion. In order to investigate the cellular regulatory mechanisms resulting from chronic TDI exposure, we focused on TGF-β1, a key factor regulated by TDI exposure. As predicted, TGF-β1 was significantly up-regulated and secreted in chronically TDI-exposed cells. In addition, SMAD2/3 was also activated considerably as it is the direct target of TGF-β1 and TGF-β1 receptors. Inhibiting TGF-β1 signaling through blocking of the TGF-β receptor attenuated EMT and cell migration in chronically TDI-exposed cells. Our results corroborate that chronic TDI exposure upregulates TGF-β1 secretion, activates TGF-β1 signal transduction, and leads to EMT and other cancer properties.

## 1. Introduction

Toluene diisocyanate (TDI) is a highly reactive, volatile chemical crucial in the manufacturing industry’s polyurethane and plastic production [1,2]. Even so, extensive polyurethane-based manufacturing industries escalate worker and nearby resident exposure to TDI [3,4]. TDI is primarily absorbed through the respiratory tract because of its volatile nature [5]. After exposure, bronchial epithelial cells undergo airway remodeling and inflammation [6]. This causes reduced lung function and respiratory diseases such as lung cancer, occupational asthma, bronchoconstriction, and bronchial irritation [7,8]. There are numerous studies on short-term acute TDI exposure but few on chronic exposure. Therefore, we aim to elucidate the effects of chronic TDI exposure on the respiratory tract and its molecular mechanisms.

Airway remodeling often results from epithelial-mesenchymal transition (EMT), or the exchange of epithelial cell characteristics for mesenchymal cell properties [9,10]. For example, intercellular junctions, apical-basal polarity, and epithelial markers are lost, while cell motility, front-rear polarity, and mesenchymal markers are gained [11]. EMT-induced cell mobility and plasticity can lead to tumorigenesis, resulting in life-threatening manifestations [12,13,14,15]. Tumorigenesis is a primary process when normal cells acquire malignant characteristics, including dedifferentiation, metastasis, angiogenesis, apoptosis evasion, and dysregulated metabolism [16]. In particular, metastasis is a hallmark cancerous property conducted through cell migration and invasion [17]. Decreasing the cell adhesion protein E-cadherin and inducing EMT prompt metastasis; therefore, activating EMT to augment cell migration and invasion can increase metastasis [18,19]. In this study, we theorized that TDI exposure causes EMT and bronchial epithelial cells’ airway remodeling, thereby leading to tumorigenesis.

Transforming growth factor-beta 1 (TGF-β1) signaling regulates various biological processes, including cell proliferation, death, and migration [20,21]. In addition, TGF-β1 has a bidirectional function, suppressing or promoting tumors [22]. TGF-β1 aberrant expression affects tumorigenesis, metastasis, and other cancer formations [23]. For instance, dysregulation induces tumorigenesis by promoting cancer stem cell homeostasis, immune evasion, and metastasis [24,25]. Therefore, we determined whether TDI exposure activates TGF-β1 signaling and influences EMT induction and metastasis.

Many studies have investigated short-term TDI exposure but not chronic exposure. Therefore, we constructed a chronic exposure model using human bronchial epithelial cells (BEAS-2B) to investigate how chronic TDI exposure affects the respiratory tract. Since TDI exposure can severely irritate the respiratory tract and cause airway remodeling, we reasoned that chronic TDI exposure would induce several cancerous properties through EMT. We aimed to determine TDI’s chronic effects on bronchial cells and the molecular mechanisms involved in pulmonary dysfunction.

## 2. Results

### 2.1. TDI Exposure Upregulates TGF-β1 as a Key Factor and Modulates Tumorigenesis-Related Genes

We measured the cytotoxicity of various TDI concentrations after one, three, and seven days to determine an appropriate transcriptome analysis concentration (Figure 1A). While the cell viability of BEAS-2B was not significantly affected on the first and third days of TDI exposure, cytotoxicity was observed at a concentration of more than 100 μM after seven days. We chose a TDI concentration that did not exhibit cytotoxicity, and transcriptome analysis was utilized to investigate TDI exposure mechanisms after one day at 50 μM and 250 μM concentrations. Differentially expressed genes (DEGs) from TDI exposure were determined through 3′ mRNA Quan-Seq. Among the 25,737 genes represented in the next-generation sequencing (NGS) analysis, 251 DEGs were identified (Appendix A, fold change ≥ 2; normalized log_2_ ≥ 5). We analyzed DEGs’ gene ontology to determine their functional groups, and our results indicated an association with apoptosis regulation, receptor activity, transcriptional regulation, extracellular matrix organization, and angiogenesis (Figure 1B and Appendix A). This signifies that TDI can cause tumorigenesis through apoptosis attenuation and angiogenesis induction in BEAS-2B cells. Therefore, we selected 44 DEGs related to tumorigenesis (Appendix A, fold change ≥1.4; normalized log_2_ ≥5). The hierarchical clustering and protein-protein interactions (PPI) of these 44 DEGs were analyzed (Figure 1C,D), confirming TGF-β1 and DEGs expression similarities and interactions. Our results suggest that TDI exposure is associated with tumorigenesis and that upregulated TGF-β1 is a prominent factor in TDI gene modulation.

### 2.2. Epithelial-Mesenchymal Transition Induces Cancerous Properties in Chronic TDI Exposure Model

TDI inhaled through the respiratory tract can injure bronchial or lung cells, and repetitive damage may induce pathological airway remodeling [26]. We constructed a chronic TDI exposure model to investigate prolonged TDI inhalation mechanisms and pathological characteristics. BEAS-2B was treated with TDI for two months for chronically TDI-exposed model construction. The cells were subcultured 20 times every 3 days. After two months of TDI exposure, cell shapes were observed through microscopy to confirm morphological changes due to acute and chronic TDI exposure (Figure 1E and Appendix A). Our results established that cellular length increased concentration-dependently when chronically exposed to TDI. EMT biomarkers, vimentin, fibronectin, and α-smooth muscle actin (α-SMA) expression levels were evaluated to determine whether cell morphological changes were caused by TDI-induced EMT (Figure 2). EMT biomarkers’ gene and protein expressions were significantly upregulated in the TDI-treated group (Figure 2A,B). Intracellular fluorescence results noted that intracellular vimentin also increased from chronic TDI exposure approximately 2-fold in a dose-dependent manner (Figure 2C,D).

EMT encourages epithelial cell plasticity; therefore, metastasis was measured through cell migration and invasion assays after chronic TDI exposure [12,27]. Cell migration was notably increased (Figure 3A,B), and cell invasion surged up to 2-fold in the 50 μM exposure group compared to the control (Figure 3C). Additionally, E-cadherin and N-cadherin expressions, which contribute to cell adhesion and motility, were analyzed [28]. E-cadherin mRNA and protein expression levels waned significantly, whereas N-cadherin expression levels rose (Figure 3D,E). These results suggest chronic TDI exposure induces EMT, thereby increasing cell migration and invasion.

### 2.3. Chronic TDI Exposure Upregulated TGF-β1 and Regulated TGF-β1/SMAD Axis Activation

TGF-β1, the major signaling pathway inducing EMT, was revealed by transcriptome analysis to have significantly upregulated expression levels after TDI treatment (Figure 1) [29,30]. TGF-β1 expression and secretion levels and its direct target, the suppressor of mothers against decapentaplegic 2/3 (SMAD2/3), were analyzed to corroborate that TDI-induced TGF-β1 signaling caused EMT and metastasis. Furthermore, to confirm whether TDI-increased TGF-β signaling resulted from the TGF-β1 ligand or the transforming growth factor-beta receptor (TGFBR), TGF-β1, TGFBR1, and TGFBR2 gene expressions were analyzed (Figure 4A). The results indicated a significant increase in TGF-β1 expression, whereas TGFBR expression did not. In addition, TGF-β1 protein expression levels increased 2.5-fold compared to the control, and TGF-β1 secretion also multiplied considerably (Figure 4B,C). When TGF-β1 was bound to TGFBR, it could phosphorylate and activate SMAD2/3 [20]. Our results verify that phosphorylated SMAD2/3 significantly increased, and total SMAD2/3 expression also slightly improved (Figure 4D–F). SMAD2/3 activity increased in the TDI 50 μM treated group, specifically. These results indicate that chronic TDI exposure increases the expression and secretion of TGF-β1 and SMAD2/3 activity.

### 2.4. TGF-β1 Signaling Inhibition Decreased TDI-Induced Cancerous Properties

SB-431542, a specific TGFBR antagonist, was co-treated in the chronic TDI exposure model to confirm TGF-β1 signaling activation, thereby inducing EMT and metastasis. Our results verify that TDI-cotreated SB-431542 significantly reduced TDI-induced cell migration and invasion (Figure 5A–C). Additionally, the TGF-β1 antagonist inhibited TDI-induced SMAD2/3 activity, which is directly regulated by TGFBR (Figure 5D,E). After SB-431542 treatment, EMT-related protein expression levels were evaluated. E-cadherin expression increased, whereas vimentin and α-SMA decreased, establishing that SB-431542 treatment attenuated EMT and metastasis induced by TDI (Figure 5F–H). These results verified that chronic TDI exposure increases TGF-β1 expression and secretion, activating SMAD2/3 and up-regulating EMT to induce cell migration and invasion.

## 3. Discussion

TDI is a highly reactive and volatile compound used in polyurethane-based manufacturing, such as packaging materials and furniture [31]. TDI is primarily absorbed through the respiratory tract and is a well-known agent for occupational asthma [5,32,33,34]. Respiratory cells exposed to TDI undergo an inflammatory response and airway remodeling, resulting in reduced pulmonary function and respiratory disease [6,8,34]. Despite these risks and increasing usage, studies on the molecular mechanisms of chronic TDI exposure are still insufficient [35]. Therefore, we aim to identify TDI-induced pathological characteristics and mechanisms through our chronic TDI exposure model.

Transcriptome analysis allows us to target genes with pronounced changes in expression after TDI exposure. In addition, genetic expression similarities and molecular interaction analyses can elucidate TDI-induced biological mechanisms [36,37]. To determine which genes and biological processes are modulated by TDI exposure, we measured total mRNA expression in BEAS-2B. Our results indicated that these DEGs are predominately involved in the initiation of cancer development, including negative apoptosis execution and angiogenesis regulation (Figure 1B). This suggests that TDI exposure can modulate cancerous properties in bronchial cells. Hierarchical clustering results revealed DEGs expression patterns, indicating a similarity in gene expression regulation from TDI treatment (Figure 1C). The PPI network’s central node is a critical factor that interacts with others in the PPI network. In our results, TGF-β1 was a key factor at the center of the networks conducting significant interactions with other DEGs (Figure 1D). We hypothesized that chronic TDI exposure could modulate gene expression associated with cancerous properties, increasing TGF-β1 expression and pathologically changing bronchial cells.

BEAS-2B cells were treated with 10 and 50 μM of TDI for two months and subcultured 20 times to create our chronically TDI-exposed model, which we used to investigate chronic TDI exposure effects on bronchial cells and their mechanisms. Cell length significantly extended when chronically exposed to 50 μM of TDI (Figure 1E). According to previous studies, epithelial cells undergoing EMT exhibit a long, spindle-like mesenchymal cell morphology [38]. Therefore, we expected these morphological changes to be related to EMT, and our results corroborated that chronic TDI exposure induced intracellular EMT biomarkers (Figure 2). 

EMT can enhance cell mobility, migration, and invasion, inducing metastasis [39]. During this process, epithelial cells are transformed into mesenchymal-like cells, cell adhesion is weakened due to modulation of adhesion molecules, and cellular mobility increases [40,41]. Our results demonstrate that chronic TDI exposure escalated BEAS-2B migration and invasion and hindered E-cadherin expression, which is responsible for cell adhesion (Figure 3). This suggests that the downregulation of cell adhesion molecules increases cell mobility and promotes metastasis. Therefore, we confirmed that chronic TDI exposure induces EMT and cell metastasis and confers cancerous properties on BEAS-2B.

TGF-β1 enacts a central role in signaling networks that regulate cell growth, differentiation, and tumorigenesis [42]. TGF-β1 is a bifunctional factor that can act as both a suppressor and a promoter depending on the tumor’s development stage [43]. Furthermore, the TGF-β1 ligand interacts with TGFBR1 and TGFBR2 via autocrine or paracrine mechanisms, activating the intracellular SMAD family and downstream signaling cascades. [44,45,46,47]. This study aimed to confirm that chronic TDI exposure activates TGF-β1 signaling, thereby inducing EMT and cancerous characteristics in BEAS-2B cells. First, we determined whether TGF-β1 signaling activation occurred through increased TGF-β1 or TGFBR expression. TGF-β1, TGFBR1, and TGFBR2 mRNA expression levels were evaluated in the chronic TDI exposure model, and only TGF-β1 expression significantly increased (Figure 4A). These results indicate that TGF-β1 signaling activation is caused by upregulated TGF-β1, not TGFBR1 or TGFBR2. In addition, increased TGF-β1 expression is also consistent with transcriptome analyses performed on TDI-exposed cells (Figure 1C,D). Chronic TDI exposure increased TGF-β1 protein expression and secretion (Figure 4B,C), and phosphorylation of the downstream factor SMAD2/3 also increased in BEAS-2B (Figure 4D–F). Thus, our results confirmed that chronic TDI exposure activated TGF-β1 signaling by upregulating TGF-β1 expression and secretion.

TGF-β1/SMAD axis activation induces EMT by increasing transcription factors, such as Snail and Slug, and decreasing E-cadherin [48]. Therefore, we expected chronic TDI exposure to activate TGF-β1 signaling and induce EMT and metastasis. As SB-431542 is a well-known TGFBR antagonist [49], cell modulations were validated after SB-431542 treatment for inhibition of TGF-β1 signal transduction. Our results demonstrated that cell migration and invasion, cancerous properties induced by chronic TDI exposure, were attenuated by SB-431542 (Figure 5A–C). Additionally, SB-431542 also inhibited the phosphorylation of the downstream TGF-β1/TGFBR signal SMAD2/3 and EMT marker expression (Figure 5D–H). This implies that TGF-β1 is a significant factor in the induction of cancerous properties from chronic TDI exposure. Furthermore, these results are consistent with recent research on TGF-β1 acting as a tumor promoter in cells undergoing tumorigenesis [50].

In this study, the molecular mechanisms of chronic TDI exposure were investigated by focusing on TGF-β1, a key factor regulated by TDI exposure. Chronic TDI exposure increased TGF-β1 expression and secretion and SMAD2/3 activity. EMT activation from chronic TDI exposure was confirmed through a decrease in molecular adhesion and an increase in mesenchymal gene expression. Moreover, our results showed that chronic TDI exposure enhanced bronchial cell migration and invasion. Inhibition of TGF-β1 signal transduction through TGFBR blockade attenuated EMT, migration, and invasion induced by TDI. Thus, our study determined how chronic TDI exposure affects the respiratory tract and is expected to further other TDI-related disease studies.

## 4. Materials and Methods

### 4.1. Chemicals and Reagents

Toluene 2,4-diisocyanate (TDI) was purchased from Junsei Chemical Co. (Tokyo, Japan). Triton X-100, sodium dodecyl sulfate (SDS), phosphatase inhibitor cocktail 2/3, protease inhibitor cocktail, 4,6-diamidino-2-phenylindole dihydrochloride (DAPI), and dimethyl sulfoxide (DMSO) were obtained from Sigma-Aldrich Chemical (St. Louis, MO, USA). SB-431542 was purchased from MedChemExpress USA (Middlesex, NJ, USA). Antibodies against anti-TGF-β1(21898-1), anti-E-cadherin (60335-1-Ig), and anti-N-cadherin (66219-1-Ig) were purchased from Proteintech (Rosemont, IL, USA). Antibodies targeting anti-p-SMAD2/3 (8828S), anti-SMAD2/3(8685S), anti-α-SMA (19245S), and anti-mouse Alexa 488 (4408S) were purchased from Cell Signaling Technology (Beverly, MA, USA). The fibronectin antibody (MA5-11981) was purchased from Thermo Fisher Scientific (Waltham, MA, USA). Anti-vimentin (sc-6260), anti-β-actin (sc-47778), HRP-conjugated anti-rabbit IgG (sc-2357), anti-rabbit CFL 488 (sc-516248), and HRP-conjugated anti-mouse IgG (sc-516102) antibodies were purchased from Santa Cruz Biotechnology (Santa Cruz, Dallas, CA, USA). 

### 4.2. Cell Culture and Treatment

Human bronchial epithelial cells (BEAS-2B) were acquired from the American Type Culture Collection (ATCC, Manassas, VA, USA). Cells were grown in a humid incubator with 5% CO_2_ at 37 °C using the Roswell Park Memorial Institute Medium (RPMI 1640, Welgene, Daegu, Republic of Korea), which contains 5% FBS (Gibco, Grand Island, NY, USA), 100 U/mL penicillin and streptomycin (Welgene, Daegu, Republic of Korea), and 1 mM sodium pyruvate (Welgene, Daegu, Republic of Korea). Cells were cultured in a 100 mm^2^ cell-culture dish and seeded at a density of 2 × 10^4^ cells/cm^2^. When confluency reached 70–80%, BEAS-2B cells were used for experimentation. Various TDI concentrations (1–500 μM) were administered for one, three, and seven days after the cells had stabilized for 24 h. For chronic TDI exposure, TDI was diluted to a final concentration of between 10 and 50 μM. After being subcultured 20 times with ongoing TDI treatment for 2 months, cells were used for further analysis. To validate TDI exposure mechanisms, 10 uM of SB-431542, a selective TGF-β receptor inhibitor, was co-treated with various TDI concentrations. 

### 4.3. Cell Viability Analysis

Cell viability was confirmed using a Quanti-Max WST-8 Cell Viability Assay Solution (WST-8 Solution, Biomax, Seoul, Republic of Korea). BEAS-2B was seeded on 96-well plates at a density of 5 × 10^3^ cells/well in 200 μL of media. Cell culture media containing TDI were replaced with RPMI 1640 medium with WST-8 solution, and cells were incubated for 45 min. After incubation, a microplate reader (Molecular Devices, San Jose, CA, USA) was used to measure each well’s absorbance at 450 nm. The treated group’s cell viability was determined by comparing the TDI-treated group with the vehicle-treated control group.

### 4.4. mRNA Extraction and 3′ mRNA Quantification Sequencing (3′ mRNA Quan-Seq)

Following the manufacturer’s instructions, total RNA was isolated using the TRIzol reagent (Life Technologies, Carlsbad, CA, USA). The isolated RNA was dissolved in RNase-free water. Agarose gel electrophoresis at 50 V for 30 min confirmed the quality of the extracted RNA. Isolated total RNA purity was determined using the Nanodrop-2000 (Thermo Fisher Scientific, Waltham, MA, USA). All sample absorbance ratios at 260 and 280 nm ranged between 1.8 and 2.0. These RNA samples were utilized for 3′ mRNA Quan-Seq or quantification real-time polymerase chain reaction (qRT-PCR). The 3′ mRNA quantification sequencing was conducted by E-Biogen Inc. (Seoul, Republic of Korea). Total RNA was isolated and prepared for library creation (Appendix A). The QuantSeq 3′ mRNA-Seq Library Prep Kit FWD for Illumina (Lexogen, Vienna, Austria) was used in accordance with the manufacturer’s instructions to build each RNA sample library. Excel-based Differentially Expressed Gene Analysis (ExDEGA, E-Biogen, Inc., Seoul, Republic of Korea) was used to quantify and visualize the results. 

Transcriptome analyses were performed adhering to previous research [51,52]. Gene ontology (GO) analysis of DEGs (fold change ≥ 2; normalized log_2_ ≥ 5) was performed. Genes were grouped and selected according to biological function using DAVID (https://david.ncifcrf.gov/, accessed on 9 March 2023). Hierarchical clustering and protein-protein interaction (PPI) analysis were conducted to identify the expression similarities and functional interaction network of selected genes (fold change ≥ 1.4; normalized log_2_ ≥ 5). Gene expression similarity was visualized using ExDEGA graphic plus 2.0 (E-Biogen, Inc., Seoul, Republic of Korea), and the interaction of proteins was predicted using STRING database 11.5 (http://string-db.org/, accessed on 9 March 2023).

### 4.5. Quantitative Real-Time Polymerase Chain Reaction (qRT-PCR)

After synthesizing 2 μg of extracted total RNA into complementary DNA (cDNA) using reverse transcriptase (ELPIS-BIOTECH, Daejeon, Republic of Korea), it was used for qRT-PCR (CFX Connect™ Real-Time PCR Detection System; Bio-Rad, Hercules, CA, USA). Synthesized cDNA, primer sets, and SYBR Green PCR Master Mix (KAPA, Wilmington, MA, USA) were used for qRT-PCR (Appendix A), and amplification was carried out under the following conditions. After initial denaturation at 95 °C for 3 min, 45 denaturation, annealing, and extension cycles were conducted (denaturation: 95 °C for 10 s; annealing: 60 °C for 10 s; extension: 72 °C for 10 s). A melting curve analysis was performed for each replicated well to evaluate PCR reaction specificity.

### 4.6. Western Blotting

Total proteins were collected using a lysis buffer containing a RIPA buffer (Bio Solution, Seoul, Republic of Korea), 2/3 phosphatase-inhibitor cocktails, and a protease inhibitor cocktail (Sigma-Aldrich, St. Louis, MO, USA). The 20 μg of protein were loaded onto 10% sodium dodecyl sulfate-polyacrylamide gel electrophoresis (SDS-PAGE), and the separated proteins were transferred to a polyvinylidene difluoride membrane (Millipore, Burlington, MA, USA). Membrane blocking was conducted using 5% skim milk at room temperature for 1 h. Next, membranes were incubated with primary and secondary antibodies in 0.5% skim milk. All primary and secondary antibodies were diluted at a ratio of 1:1000 and 1:2000, respectively. ECL Plus western-blotting detection reagent (Amersham Bioscience, Buckinghamshire, UK) was used for detecting target proteins. Images were taken with the Bio-Rad ChemiDoc XRS System (Hercules, CA, USA). To normalize the data, β-actin was employed as a loading control, and data quantification was conducted using Quantity One imaging software (Bio-Rad, Hercules, CA, USA).

### 4.7. Cell Migration and Invasion Assays

Cell migration assays were conducted using Culture-insert 2 wells (ibidi, Munich, Germany) to form equal gaps between experimental groups. After inserts were adhered to the 12-well plate, cells were seeded with media containing TDI in each culture insert and incubated until cell confluency reached 90%. When the cell reached proper confluence, the insert was carefully removed, and the media was replaced with serum-free media. After 24 h, cell migration was observed through an inverted microscope (Leica Microsystems, Wetzlar, Germany). The cell migration area was quantified by ImageJ software version 1.4.3 (National Institutes of Health, Bethesda, MD, USA).

Cell invasion assays were performed using a 96-well cell invasion assay kit (Cell Biolabs Inc., San Diego, CA, USA) following the manufacturer’s instructions. Cells were seeded with serum-free media into the upper chambers at a density of 2 × 10^4^ cells/well. The serum-containing medium was added to the lower chamber as an attractant, and the plate was incubated at 37 °C for 24 h. Invaded cells were separated by Cell Detachment Solution and lysed using a lysis buffer. Cell lysates were stained with CyQuant^®^ GR dye solution, and fluorescence was detected by an Infinite F200 Pro multimode microplate reader (Tecan, Männedorf, Switzerland) at 485/520 nm. Relative migrated area and invasion fluorescence were calculated and compared to the controls.

### 4.8. Immunocytochemistry (ICC) Staining

Before the experiments, the 6-well plates were pre-coated with poly-L-lysine (Sigma-Aldrich Chemical, St. Louis, MO, USA). TDI-exposed cells were fixed with 4% formaldehyde (Sigma-Aldrich, St. Louis, MO, USA) for 10 min. Next, cell permeabilization for staining was conducted using 0.25% TritonX-100 (Sigma-Aldrich Chemical, St. Louis, MO, USA) for 10 min. The 1% bovine serum albumin (Sigma-Aldrich, St. Louis, MO, USA) was used for blocking, and primary antibodies were added to cells. Then, cells were treated with secondary antibodies and sequentially stained with DAPI. Fluorescent images were acquired by a confocal microscope (Carl Zeiss, Oberkochen, Germany), and ImageJ software was used for quantification.

### 4.9. Enzyme-Linked Immunosorbent Assay (ELISA)

To analyze secreted TGF-β1 volume, the supernatants were gathered after TDI exposure. TGF-β1 quantities in the obtained supernatant were measured using the TGF-β1 Human/Mouse Uncoated ELISA Kit (Thermo Fisher Scientific, Waltham, MA, USA). The assay was conducted following the manufacturer’s instructions, and protein level quantifications were calculated and compared to the control.

### 4.10. Statistical Analysis

All experiments were carried out three times, and experimental data were expressed as the mean ± standard error of the mean (SEM). The statistical significance of the results was determined by the one-way ANOVA test with Tukey’s post-analysis. *p*-values < 0.05 were regarded as statistically significant.

## Figures and Tables

**Figure 1 ijms-24-06157-f001:**
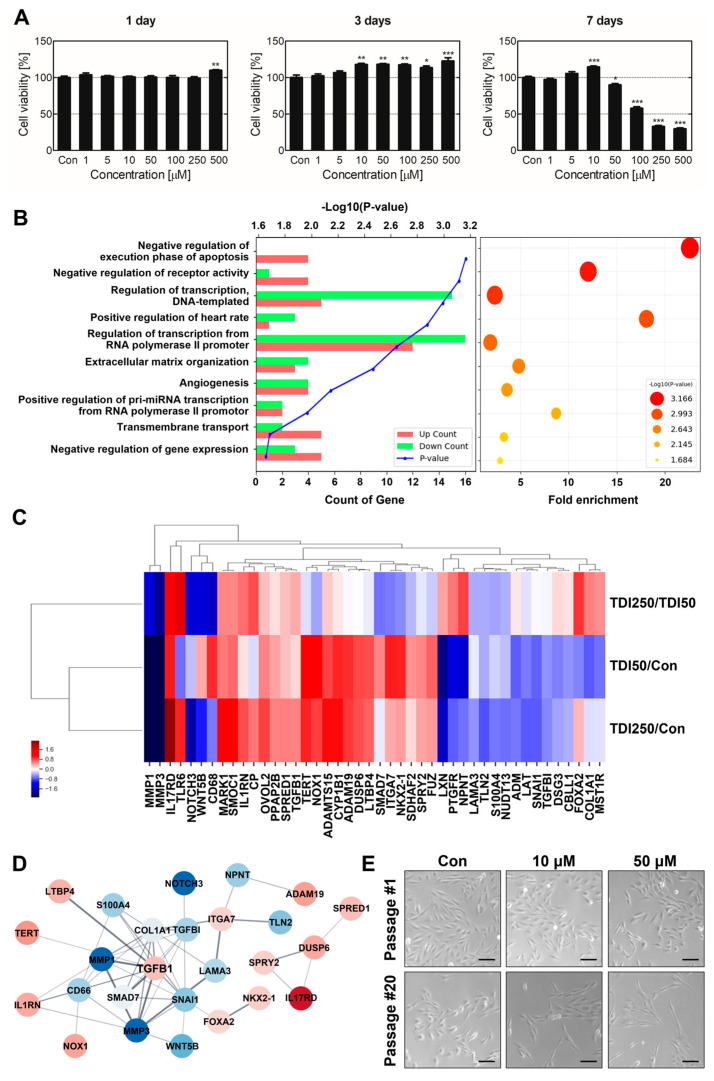
BEAS-2B transcriptional and morphological alteration after TDI treatment. (**A**) Cell viability was measured for TDI-exposed BEAS-2B cells after 1, 3, and 7 days. (**B**) After exposure to 50 μM and 250 μM of TDI for 24 h, gene ontology (GO) and gene set enrichment analyses were performed using 251 differentially expressed genes (≥2 folds, ≥5 normalized data values). (**C**) Hierarchical clustering was performed using a fold change value of 44 DEGs related to cancerous properties. Red-blue colors indicate relative changes in TDI50/Con and TDI250/Con gene expression. Similar expression patterns determined gene distance. (**D**) For 44 DEGs, a PPI analysis was conducted using the string database. Significantly increased or decreased genes are represented by red and blue nodes, respectively. Connected nodes have a significant co-expression relationship demonstrated by bold lines. (**E**) BEAS-2B morphological changes during short-term and chronic TDI exposures were observed by microscope. Chronic TDI exposure led to morphological alterations, including cell elongation and shape change. Every experiment was performed over three times. * *p* < 0.05; ** *p* < 0.01; *** *p* < 0.001 compared to the control group; TDI250: 250 μM of TDI; TDI50: 50 μM of TDI; Con: control; Scale bars: 50 µm.

**Figure 2 ijms-24-06157-f002:**
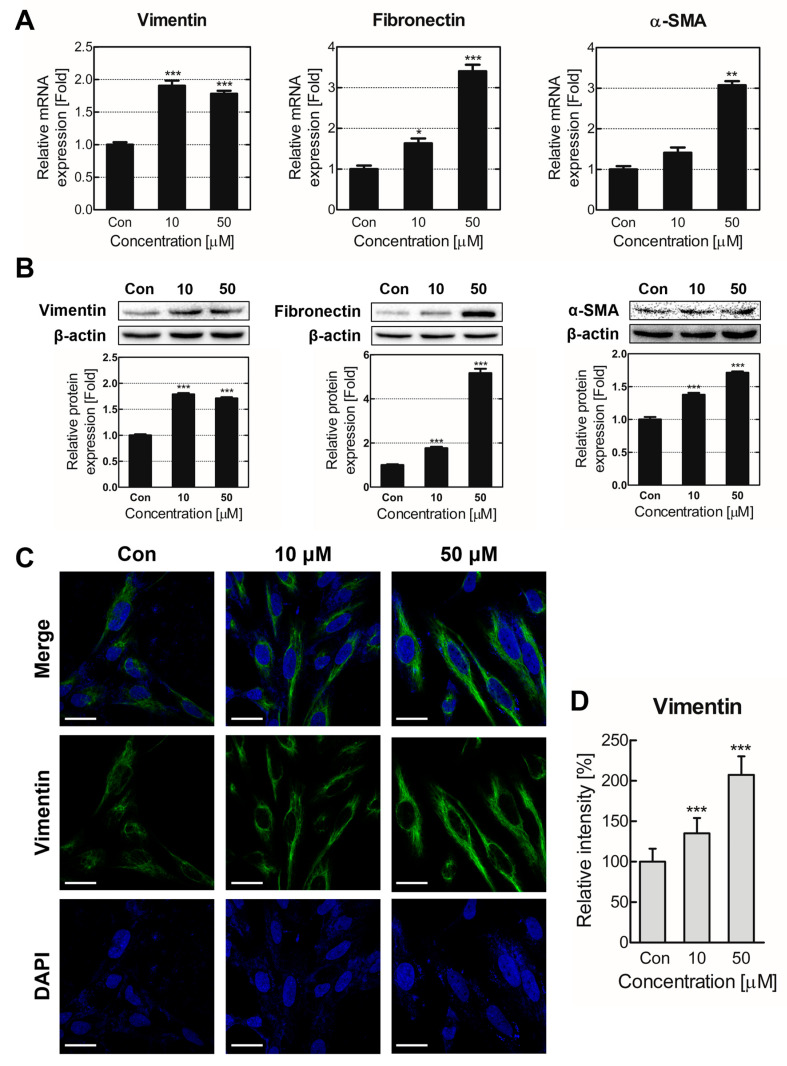
Epithelial-mesenchymal transition (EMT) gene expression increased from chronic TDI exposure. EMT marker (vimentin, fibronectin, and a-SMA) expression levels were confirmed through the chronically TDI-exposed model. (**A**) Vimentin, fibronectin, and α-SMA gene expression levels were analyzed through qRT-PCR. (**B**) Vimentin, fibronectin, and α-SMA protein expression levels were measured. (**C**) Alexa 488 and DAPI labeled the intracellular vimentin and nucleus, respectively. (**D**) Intracellular vimentin was quantified through Image J software. Every experiment was performed over three times. * *p* < 0.05; ** *p* < 0.01; *** *p* < 0.001 compared to the control group; Con: control; Scale bars: 20 μm.

**Figure 3 ijms-24-06157-f003:**
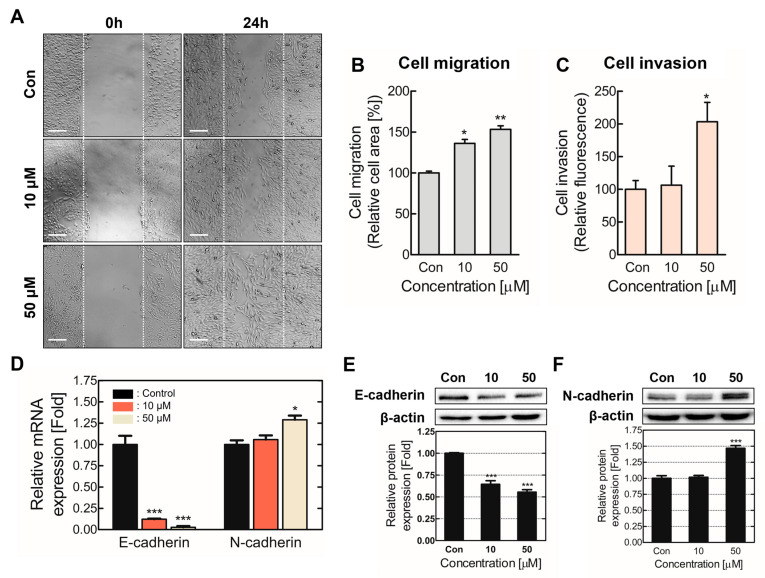
Chronic TDI exposure increased cell migration and invasion. (**A**) Cell migration in the chronically TDI-exposed model was observed through microscopy. (**B**) Migration quantification is represented in the bar graph. (**C**) Chronically TDI-exposed BEAS-2B’s basement membrane permeability was analyzed through a cell invasion assay. E-cadherin and N-cadherin (**D**) intracellular genes and (**E**,**F**) protein levels were detected. Cell adhesion molecules involved in cell migration and invasion were evaluated in the chronic TDI exposure model. Every experiment was performed over three times. * *p* < 0.05; ** *p* < 0.01; *** *p* < 0.001 compared to the control group; Con: control; Scale bars: 200 µm.

**Figure 4 ijms-24-06157-f004:**
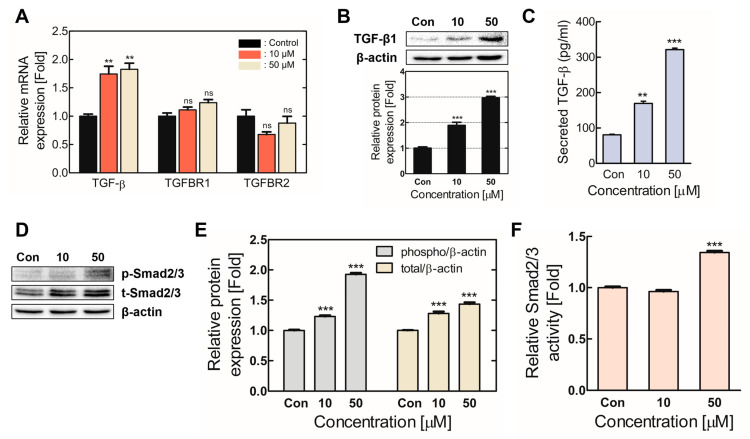
Chronic TDI exposure activated the TGF-β1/SMAD axis. TGF-β1/SMAD signaling factor expression and activity levels were confirmed in the chronically TDI-exposed model. (**A**) TGF-β1, TGFBR1, and TGFBR2 gene expression levels were evaluated. TGF-β1 (**B**) protein expression and (**C**) secretion levels were detected. (**D**,**E**) Phosphorylated- and total-SMAD2/3 protein levels were assessed. (**F**) SMAD2/3 activity was evaluated through phosphorylated-SMAD/total-SMAD quantification. Every experiment was performed over three times. ns: not significance; ** *p* < 0.01; *** *p* < 0.001 compared to the control group; Con: control; 10: 10 μM of TDI; 50: 50 μM of TDI.

**Figure 5 ijms-24-06157-f005:**
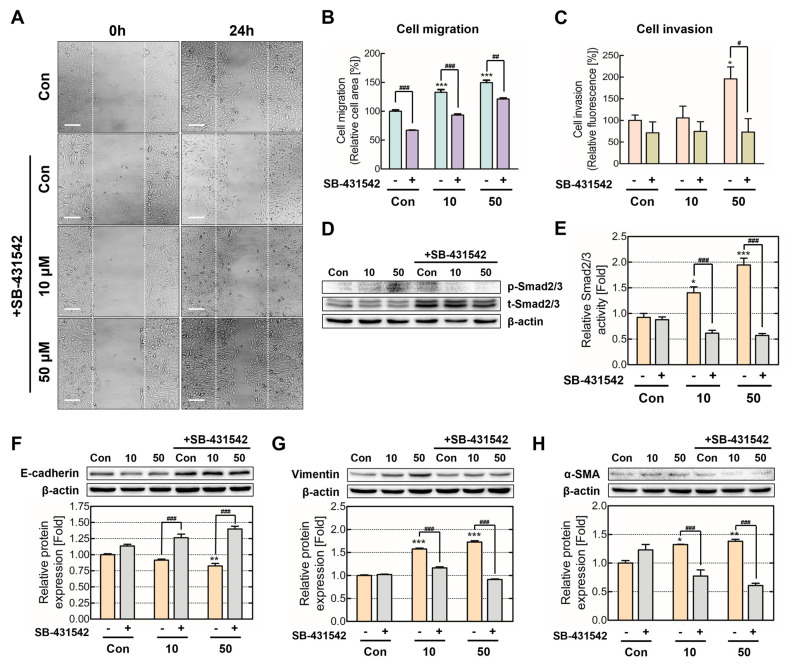
Cell metastasis and EMT attenuation in chronic TDI exposure via TGF-β antagonists. To validate TGF-β1/SMAD axis regulatory effects, the chronic TDI exposure model was co-treated with TDI and the TGFBR-specific antagonist SB-431542. (**A**) Migration was observed through microscopy after TDI and TGF-β antagonist co-treatment. (**B**) The bar graph represents quantified values. (**C**) Cells basement membrane permeability was measured using a fluorescence reader after TDI and TGF-β antagonist co-treatment. (**D**) p-SMAD2/3 and SMAD2/3 protein expression levels were measured. (**E**) SMAD2/3 activity was calculated through phosphorylated-SMAD/total-SMAD quantification. (**F**) E-cadherin, (**G**) vimentin, and (**H**) a-SMA protein expression levels were measured. Every experiment was performed over three times. * *p* < 0.05; ** *p* < 0.01; *** *p* < 0.001 compared to the control group; # *p* < 0.05; ## *p* < 0.01; ### *p* < 0.001 compared to SB-431542 untreated group; Con: control; 10: 10 μM of TDI; 50: 50 μM of TDI; Scale bars: 200 µm.

## Data Availability

Not applicable.

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
