# Peer review of "Chronic Exposure to TDI Induces Cell Migration and Invasion via TGF-β1 Signal Transduction"

_ijms, 2023, doi:10.3390/ijms24076157_

Round 1

Reviewer 1 Report

The manuscript entitled “Chronic Exposure to TDI Induces Cell Migration and Invasion 2 via TGF-β1 Signal Transduction” written by Han et al. aims to identify the molecular mechanisms of bronchial epithelial cells after chronic TDI exposure. This study is interesting and authors have provided information very well. However, I have concerns regarding following:

·      Authors have explained well regarding the TDI exposure in cells. However, it will improve the quality if they can include some data from in vivo study as well and please mention how many times experiments were repeated to confirm the results.

·      Line 71: Please rephrase this line and mention the cells regarding which authors are discussing here.

·      Line 76-86: Please provide the percentage or number of total upregulated and downregulated genes as well as commonly expressed genes in all groups. Authors can include bar graph or venn diagram to explain this.

·      For transcriptome analysis authors used 50 μM and 250 μM concentration of TDI. However, for remaining experiments concentration used was 10 and 50 μM. Why concentration of TDI is different for this study ?

·      Please mention about the SB-431542 in section 4.1, from where it was purchased.

·      Line 303: Please recheck this 5 × 104 cells/ml density.

·      Section 4.4: Did authors check the quality as well as RIN values of RNA before sequencing?

·      Please include the concentration of RNA used for sequencing as well as library preparation.

Reviewer 2 Report

The manuscript titled „Chronic Exposure to TDI Induces Cell Migration and Invasion via TGF-β1 Signal Transduction” by Dong-Hee Han, Min Kyoung Shin, Jin Wook Oh, Junha Lee, Jung-Suk Sung, Min Kim decribes the influence of chronic exposure to the chemical, toluene diisocyanate (TDI), on bronchial epithelial cells. Authors decribe the TGF- β1 induction and epithelial-mesenchymal transition, showing the inhibition or induction of specific markers of this process.

The manuscript is well organized and the data are clearly described.

There are some minor concerns. Generally, the English language needs some stylistic improvement, e.g. Especially, in Introduction, some thought are repeated several times.

1.       The second and fourth sentences of the Abstract are not clear

2.       The title of section 2.1 is not clear

3.       Instead of „on seven days” use e.g. „after seven days” or „on seventh day” (line 72)

4.       Is the x-axis title in Figure 1B, second part, correct?

5.       Are genes in Figure 1C clustered based of fold changes?

6.       In Figure 1D, there is „Con” for control but in other parts of this figure it is „C”

7.       In Figure 1B title, it is unclear „after TDI exposure”

8.       The sentence in lines 107-108 is unclear

9.       Are the results in Figure 2A for qRT-PCR?

10.   Authors could state the numer of biological repeats in every description to figures

11.   Is it possible to show the cel lenght change as a plot?

12.   The sentence in line 297 is not clear

13.   Was the inhibitor SB-431542 also supplied for 20 passages?

14.   The bioinformatic analysis could be described in more detail

15.   The sequences of primers used in qRT-PCR could be supplied in e.g. as supplementary data

16.   The authors could supply the dilutions of antibodies used

17.   Is the cell migration assay decribed correctly in Materials and Methods, as compared to the Figure 3A?

Generally, the authors decribe the influence of chronic exposure of TDI on the cells. In Discussion section, it seems that it lacks a kind of comparision to the other published results treating the chronic but as well as acute exposure to TDI.
